# Applications of Wearable Technology in a Real-Life Setting in People with Knee Osteoarthritis: A Systematic Scoping Review

**DOI:** 10.3390/jcm10235645

**Published:** 2021-11-30

**Authors:** Tomasz Cudejko, Kate Button, Jake Willott, Mohammad Al-Amri

**Affiliations:** School of Healthcare Sciences, College of Biomedical and Life Sciences, Cardiff University, College House, King George V Drive East, Heath Park, Cardiff CF14 4EP, UK; buttonk@cardiff.ac.uk (K.B.); WillottJ@cardiff.ac.uk (J.W.); al-amrim@cardiff.ac.uk (M.A.-A.)

**Keywords:** joint disease, wearable devices, e-health, free-living, artificial intelligence

## Abstract

With the growing number of people affected by osteoarthritis, wearable technology may enable the provision of care outside a traditional clinical setting and thus transform how healthcare is delivered for this patient group. Here, we mapped the available empirical evidence on the utilization of wearable technology in a real-world setting in people with knee osteoarthritis. From an analysis of 68 studies, we found that the use of accelerometers for physical activity assessment is the most prevalent mode of use of wearable technology in this population. We identify low technical complexity and cost, ability to connect with a healthcare professional, and consistency in the analysis of the data as the most critical facilitators for the feasibility of using wearable technology in a real-world setting. To fully realize the clinical potential of wearable technology for people with knee osteoarthritis, this review highlights the need for more research employing wearables for information sharing and treatment, increased inter-study consistency through standardization and improved reporting, and increased representation of vulnerable populations.

## 1. Introduction

Osteoarthritis (OA) is the most prevalent chronic joint disease, with the knee being the most commonly affected joint [1,2]. It is estimated that about 250 million people worldwide are affected by OA [3]. OA is characterized by irreversible articular cartilage degeneration, chronic pain, and disability, and 13% of OA patients are forced into early retirement, with an average loss of eight working years per patient [4]. The medical cost of OA in various high-income countries accounts for between 1% and 2.5% of their gross domestic product [5]. As the number of people affected with OA is likely to increase due to aging populations and the obesity epidemic [6], so too will the demand on healthcare resources. In addition, the COVID-19 pandemic has had a notable impact on primary care services which have had to adapt to different methods of service provision to limit the number of face-to-face appointments with patients [7]. Concurrently, there has been a rise in the research and development of wearable technology for healthcare applications for this population [8,9], which could facilitate the treatment and monitoring of patients outside of a clinical setting.

Wearable technology can be defined as any wearable and portable device with the ability to detect or record any health indicators and/or as a device that allows information sharing and/or treatment to the wearer. Wearable technology has had a significant influence on the fitness industry, with mobile phones, apps, and wearable sensors in widespread use by citizens [10]. Moreover, innovations such as artificial intelligence (AI) are finding new uses in patient self-care and clinical management, enabling the growth of personalized medicine [11]. Thus, the widespread adoption of wearable technology in a real-world setting has the potential to change how healthcare is provided. 

The transformation of the wearable technology landscape is evidenced and synthesized in numerous focused review articles. Matthew-Maich et al. provided guidelines for designing, implementing, and evaluating mobile health technologies for the management of chronic conditions in elderly people [12]. Another review summarized the usage of accelerometers for monitoring of physical activity and sedentary behavior and provided practical considerations for their adoption in orthopaedics [13]. Kvedar et al. examined the values of telehealth and telemedicine to patients and professionals [14]. Within the OA research field however, knowledge regarding current utilization of wearable technology seems limited. Summarizing and disseminating information from peer reviewed, published research can stimulate and facilitate knowledge transfer between policymakers and relevant stakeholders, and thus guide the further development of wearable technology. 

Kobsar et al. conducted a scoping review examining use of wearable sensors in people with lower-limb OA but focused primarily on gait analysis, rather than a free-living assessment broadly [8]. Another scoping review investigated wearable sensors for the assessment of outcomes following knee joint replacement surgery [9]. In their narrative review, Sliepen et al. provided an overview of physical activity monitors in knee and hip OA with a focus on technical specifications [15]. Thus, the way wearable technology has been deployed in knee OA research in a real-world setting remains unexamined. In addition, to our knowledge, the facilitators and barriers to the feasibility of using wearables in a real-world setting in people with knee OA have not been determined. A better understanding of such facilitators and barriers will shed light on users’ engagement with wearables and provide guidance for planning and designing research in the field. Mapping the breadth of this field will benefit clinicians, industry, and policy makers as they are involved in developing best practices for use of wearables and facilitating understanding how this technology can be of benefit to people with knee OA. 

Therefore, the objective of this scoping review is to identify and map existing evidence describing the current utilization of wearable technology in a real-world setting in people with knee OA. 

## 2. Materials and Methods

We employed the scoping review methodology developed by Arksey and O’Malley [16], which includes five stages: (1) identification of a research question(s), (2) identification of relevant studies, (3) study selection, (4) charting the data, and (5) data synthesis. Reporting adheres to the Preferred Reporting Items for Systematic Review and Meta-Analyses (PRISMA) Extension for Scoping Reviews [17] (Appendix A). The protocol, with its predefined search strategy and criteria for eligibility, study selection, data extraction, and analysis was registered within the Open Science Framework prior to the commencement of the study (https://osf.io/3fh7m accessed on 25 November 2021). A deviation from the protocol was that we did not consult relevant stakeholders, which is also recommended in the scoping review methodology developed by Arksey and O’Malley [16]. The number of studies involved, the broad topic range, and the lack of a qualitative analysis of identified papers defined our approach as a scoping review and differentiated it from a systematic review [16]. This review provides valuable answers on the feasibility (i.e., does any evidence in the literature exists), relevance (i.e., has a systematic review already been done), and time needed (i.e., the volume of the evidence) to conduct a systematic review.

### 2.1. Research Questions

The objective of this work was achieved via addressing the following research questions:(1) what type of wearable technology is used in a real-world setting in people with knee OA, (2) what are the applications of wearable technology in people with knee OA, and (3) what are the facilitators and barriers to feasibility of using wearable technology in a real-world setting in people with knee OA.

### 2.2. Eligibility Criteria

We included full-length journal articles with any type of study design theoretically possible, i.e., randomized controlled trials, non-randomized trials, cohort studies, case-control studies, case series. We also included qualitative studies that served as the basis for the evaluation of wearable technology, provided the study participants used wearable technology in a real-world scenario. We excluded secondary types of evidence such as reviews, comments, editorials, dissertations, conference proceedings, etc. Included studies needed to be conducted in a real-world setting (e.g., residence or community setting) where participants used a form of wearable technology. Studies conducted explicitly in a university laboratory setting, such as validation studies of wearable technology, were excluded. We defined wearable technology as an electronic device that is worn close to and/or on the surface of the skin, where it collects information concerning any health indicators or which allows information sharing/treatment to/from/for the wearer. Participants were adults (older than 18 years of age) with knee OA. We excluded studies with participants scheduled for/or after any form of joint surgical procedure, with other forms of arthritis (e.g., rheumatoid arthritis), and, in case of mixed populations, studies that do not differentiate/stratify data results between the knee OA population and those having other condition. 

### 2.3. Information Sources and Search

A systematic search for published evidence was conducted in MEDLINE (PubMed) and then customized for the CINAHL, Scopus, Web of Science, EMBASE, IEEE Digital Library, and Cochrane Library. Additionally, secondary searches included: ACM Digital Library, Google Scholar, and reference lists of included studies. All databases were searched from inception until 3 June 2021. The search strategy identified records that contained at least one search term in each of the following themes: joint area, disease, and wearable technology. Included studies could be published in English, Polish, Arabic, or Spanish. Search strategy and results for each database are presented in Appendix A.

### 2.4. Study Selection

Screening of manuscript titles and abstracts was conducted by two independent, blinded reviewers (T.C. and J.W.). The search results from all databases were combined and duplicate studies were removed. Prior to screening, reviewers worked through a pilot exercise of example titles and abstracts. Discrepancies between the reviewers were resolved by a discussion. The third reviewer (M.A.A.) was consulted if no consensus could be reached. The full text was obtained for all studies that passed screening by title and abstract. The first author (T.C.) assessed all full-text articles, while the second and the last author (K.B., M.A.A.) assessed half of the full-text articles for eligibility, and nominations for exclusion were discussed and agreed on by the authors. To ensure an independent review process, screening of manuscript titles and abstracts as well full texts was conducted through the use of the Rayyan systematic review web app [18]. In case of full text missing, we contacted the first author by e-mail to obtain the full text. Authors were given one week to respond and sent a reminder if needed.

### 2.5. Data Extraction

Following the screening process, data from all included full-text manuscripts were collected within a data extraction spreadsheet (Excel, Microsoft Corporation, Redmond, WA, USA) by the first author (T.C.) and then verified by the last author (M.A.A.). Any discrepancies were resolved by consensus, and, if needed, a third reviewer was consulted (K.B.). Data extracted from the manuscripts included general publications details (authors, year of publication, language, etc.), information about the study (design, sample size, etc.), and information about the studied population (gender, body mass index (BMI), age, disease, etc.). In addition, specific information addressing each research question was extracted. A detailed data extraction template is presented in Appendix A.

### 2.6. Data Synthesis

We recorded the data extracted from each study in tables and summarized it by using visualizations such as bar graphs, histograms, and pie charts, using Tableau Public 21.2 (TABLEAU SOFTWARE, LLC). Specifically, we first synthesized general study and population details by visualizing the number of studies by year of publication, design, sample size, country, mean age, mean Body Mass Index (BMI) group, and ethnicity of participants. To answer what type of wearable technology is used in people with knee OA in a real-world setting, we visualized the following metrics: type of wearable technology, body location of use/attachment, and duration of use. To depict applications of wearable technology in knee OA, we visualized the aims of wearable technology and specific metrics collected with or transferred to wearable technology from the wearer. Finally, to demonstrate facilitators and barriers to the feasibility of using wearable technology in a real-world setting in people with knee OA, we collected and synthesized data on the following metrics: facilitators and barriers, user experience, adherence to and adverse events related to using wearable technology, instructions provide to participants how to use wearable technology, how data obtained from wearable technology was analyzed, and reliability and validity of wearable technology.

## 3. Results

The database searches resulted in 3656 records. Following duplicate removal, 2487 titles and abstracts were assessed for eligibility. Of these, 2219 records did not meet the inclusion criteria. Subsequently, 266 full-text articles were assessed for eligibility, and 68 studies [19,20,21,22,23,24,25,26,27,28,29,30,31,32,33,34,35,36,37,38,39,40,41,42,43,44,45,46,47,48,49,50,51,52,53,54,55,56,57,58,59,60,61,62,63,64,65,66,67,68,69,70,71,72,73,74,75,76,77,78,79,80,81,82,83,84,85,86] were ultimately included in the review (Figure 1).

### 3.1. General Study and Population Details

In total, 35.3% of the studies were published in the last two full years included in the review (Figure 2A). Retrospective cohort (19.1%), RCT (14.7%) and cross-sectional (11.8%) were the most reported study designs (Figure 2B). The USA was the country where the majority of the studies were conducted (51.5%), followed by Canada (11.8%) and the UK (10.3%) (Figure 2C). 35.3% of studies recruited ≤50 participants with knee OA (Figure 2D). Female participants constituted the majority of study subjects in most of the studies (Figure 2F). Most of the studies recruited people with knee OA aged in the range of 60 to 70 years old (Figure 2E), and in the BMI range of 28 to 32 kg/m^2^ (Figure 2G). 58.8% of the studies did not report the ethnic background of participants (Figure 2H).

### 3.2. Wearable Technology

Accelerometers were the most used type of wearable technology, accounting for 63% of wearables reported in the studies (Figure 3A). The reported wearable technology was mostly attached in the regions of the pelvis (19.8%), hip (16.1%), and mid-thigh (13.6%), while hand-held types of wearable technology accounted for 13.6% of all reported technology types (Figure 3B). In 69.1% of studies, participants were instructed to use wearable technology for less than one week (Figure 3C).

### 3.3. Applications of Wearable Technology

We identified three main aims of wearable technology use, with outcome assessment being the most common application (72%) (Figure 3D). Specific data metrics recorded/transferred from/to the wearer are organized according to the type of wearable technology and presented in Figure 4. Time spent in moderate to vigorous or sedentary physical activity and the number of steps per day were the metrics collected most often (Figure 4).

### 3.4. Facilitators and Barriers to Feasibility of Using Wearable Technology

In total, 82.3% of studies did not reflect on any facilitators or barriers to the feasibility of using wearable technology in a real-world setting in people with knee OA (Figure 5A). Ease of use was the most reported facilitator. However, we observed that the majority of studies did not report on user experience (Figure 5B) and adherence to using wearable technology (Figure 5C), did not report whether they provided instructions to participants on how to use wearable technology (Figure 5D) and did not report adverse events related to using wearable technology (Figure 5E). One study involved the public in wearable technology design [35].

Another mentioned facilitator was a consensus on how data obtained from wearable technology should be analyzed. We determined that 42% of the studies collecting sensor data with wearable technology did not report how they analyzed raw data, and out of those who reported, half used a commercially available producer software, whereas the other half own processing algorithms (Figure 5F). In addition, we observed that the validity and reliability of the wearable technology were not reported in 61.4 and 66.6% of studies, respectively (Figure 5G,H). Likewise, 80.7% of studies did not include raw sensor data collected with wearable technology, and none of the studies accompany their publication with publicly available software/algorithms for data analysis (Appendix A).

Finally, low cost and longer duration of battery time of wearable technology were also commonly reported by facilitators. However, we identified that 97 and 95.6% of studies did not report the price and battery time of wearable technology used in their studies (Appendix A).

## 4. Discussion

The objective of this review was to identify and map the available evidence describing current utilization of wearable technology in a real-world setting in people with knee OA. We presented an overview of type of wearable technology used in a real-world setting in people with knee OA. We also determined what are the applications of wearable technology in this population. Finally, we identified the most critical facilitators and barriers to the feasibility of using wearable technology in a real-world setting in people with knee OA that may enhance the generalized adoption of wearable technology in this patient group. 

The results of this review reflect the very recent growth in the use of wearable technology in a real-world setting in people with knee OA, with 35.3% of the studies published in the last two full years included in the review. Variability in the study designs reported represents the variety of the context that wearable technology was used in. Most studies recruited relatively young participants (62–66 years of age) and in the overweight/obese BMI category (28–32 kg/m^2^), which is in line with a review by Kobsar et al. [8], but questions the generalizability of the results to older populations with a more severe OA stage. We did not collect data on radiological severity of knee OA, which could have provided more insights regarding the generalizability of the results to people with a more severe knee OA diseases stage and thus can be a topic for future research. Although the majority of the studies were limited to less than 50 participants, cohort studies such as the Osteoarthritis Initiative (n > 1000) [70] or the Multicentre Osteoarthritis Study (n > 400) [61] provide convincing evidence that wearable technology can be used in a large scale in a real-world setting in people with knee OA. Few studies collected information on race/ethnicity, despite its considerable significance for outcomes and disparities [87]. Wearable technology presents an opportunity to reduce workforce, financial, and geographic barriers [88]. The relative scarcity of studies in vulnerable populations suggests the need for improved representation to ensure that wearable technology is designed for use in all populations and that it does not exacerbate disparities.

### 4.1. Applications of Wearable Technology in Knee OA

Currently, wearable technology is used for outcome assessment, information sharing, and treatment. 

An accelerometer was the most common wearable technology used for outcome assessment. Although the use of an accelerometer allows researchers and clinicians the ability to track a broad scope of outcomes, its current use in a real-world setting is limited to assessing physical activity metrics. Nevertheless, accelerometers provide greater objectivity in assessing physical activity than patient-reported outcomes which are subject to the ceiling effect and recall bias [89]. Moreover, accelerometers can measure the four dimensions of physical activity: frequency, intensity, time, and type, but an inability to measure all activities with equal accuracy has been suggested as their current limitation [15]. In addition, data processing methods for accelerometers are not generalizable to all populations because they are based on specific thresholds subject to differences in movement patterns and walking speed between populations [59]. An overwhelming number of studies placed an accelerometer near the center of mass. However, the accelerometer anatomical placement should be chosen appropriately based on the research question as it influences their characteristics, user acceptance, engineering requirements, and data processing choices [90].

In health care, sensor data such as accelerometry of gyroscope data would be even more relevant if combined with simultaneously collected patient-reported outcomes. This would enable adding context to the sensor metrics, monitoring of symptoms, and can aid clinical decision making and empower patients. This review demonstrates the ability to collect patient-reported outcomes such as pain intensity, KOOS and WOMAC by using tablets, smartwatches, and smartphones. Most studies were observational in nature and represented outcome assessment in non-experimental conditions with the aim to establish associations between wearable technology-derived health metrics and other health outcomes. Nevertheless, the overall variability of the metrics reported, and in some cases inconsistencies in methods and data processing choices, calls into question the current ability of wearable technology-derived outcomes to inform health prevention and management efforts. 

Current research in user-related information delivery via wearable technology is primarily concerned with the provision of feedback, real-time user insight, and recommendations. Tablets, smartwatches, or smartphones were utilized in 15 studies for information sharing, such as reminding participants about using another wearable technology, providing information/advice about the importance of participation in physical activity/exercise, or connecting with a health professional or with other study participants. Examples from other areas demonstrate that wearable technologies incorporating an information-sharing function within the device can enhance communication between patients, professionals, and caregivers, provide more opportunities to express feelings, increase connectedness and caregiver support and improve advanced care planning [91]. Smartphones, tablets and/or smartwatches are accessible and can be delivered conveniently and easily to the target audience. However, these devices may not be suitable for all people with knee OA, potentially because of age, preference, comorbidities, and/or severity of illness. Manini et al. observed that App interface customization was a recurring theme throughout the focus group discussions, pointing to the need to accommodate potential hearing and visual impairments and to users’ individual needs [92]. Therefore, using wearable technologies for information sharing must be tailored to the needs, preferences, and conditions of patients and be supported by face-to-face modes in the intervention package. 

We found 11 studies that used wearable technologies, such as ultrasound, PES, PEMF, and or TENS for the treatment of people with knee OA in a real-world setting. Many people with knee OA may live at a distance from health services and continue to be in the workforce with limited time to attend treatment sessions. Thus, such wearable devices have the potential to be an important component in models of care by offering a sustainable opportunity to improve patient outcomes via the enhanced delivery of self-management tools and increased access to best practices and continuity of care [93]. Although some reviews suggested ultrasound and PEMF to be effective in reducing pain and improving physical function in people with knee OA [94,95], it should be noted that current international clinical guidelines for the management of knee OA do not recommend such devices to be considered a core treatment for knee OA [96,97].

### 4.2. Facilitators and Barriers to Feasibility of Using Wearable Technology in a Real-World Setting

The possibility of enriching people with knee OA with wearable technology to manage their own health and facilitating clinicians with superior methods of monitoring highlights the potential of such devices [98]. With technology improving drastically, we can expect such devices to be widely adopted in patient care in the future. A prerequisite for this is a better understanding of the facilitators and barriers to the feasibility of using wearable technology in a real-world setting. Our review demonstrates, however, that these aspects have not been addressed in most of the studies. 

The user feedback reviews report that initial user enthusiasm on wearable devices is often lost because of technical complexities, price, data quality concerns, and unclear end-user needs [99]. Indeed, we observed low technical complexity was the most reported facilitator. Data on user experience and engagement is crucial for the widespread adoption of such devices for patient self-management. Very few studies, however, evaluated user experience, adverse events related to the use of wearable technology, nor did they report whether they provided instructions to participants on how to use the wearable technology. Simple interface, technical support, and clear instructions are needed to tackle the technological barriers, which is consistent with other studies [100,101].

Low cost of wearable technology or subsidization of costs by health insurance was another facilitator. Only two studies reported the price of wearable technology used in their studies with a reported price of $4400 to $6800 for ultrasound devices [26,33]. Sliepen et al. reported that the financial burden varies widely between activity monitors for knee and hip OA patients, ranging from €25 to €4500 [15], demonstrating a difference in the price of wearables depending on their applications (data collection vs. treatment). Authors also reported that the need for and cost of required software varies greatly between the devices, which makes it difficult to understand the overall cost of the wearable technologies.

Establishment and disclosure of specific threshold for estimation of physical -activity metrics derived from accelerometers and consensus on how this data should be analyzed was another facilitator. Collins et al. observed that specific thresholds to measure PAs derived from hip-worn accelerometers should not be used to measure PA by wrist-worn accelerometers in adults with knee OA [59]. Our review demonstrated variability and lack of consensus regarding processing choices for sensor data from wearable devices. We additionally show that 50% of included studies used wearable technology, which was validated/tested for reliability on the non-knee OA population. This is consistent with a recent meta-analysis demonstrating that IMU devices are generally validated with highly heterogeneous protocols, in different populations, and against varying criteria methods [102]. The validation of wearable technologies explicitly in healthy individuals could lead to an erroneous reflection of the device’s true validity, as knee OA leads to abnormalities in gait dynamics and increased energy expenditure during gait compared to healthy participants [103]. Moreover, validation studies are often conducted in a laboratory setting, which represents an artificially constrained environment deviating from a real-world setting, thereby confounding locomotor parameters compared to habitual gait performance [104].

Another facilitator was the ability of technology to connect/provide feedback to a health professional or the ability to receive feedback in the form of visualizations of patient’s data metrics or information about the disease. Participants emphasized the importance of using easily understandable and interpretable health metrics on which counseling can be based. If wearable technology provides feedback, it is important that it has respectful automation (appropriate frequency of questionnaires) and tone of feedback. Beukenhorst et al. reported that too frequent reminders about the importance of participation in physical activity sent via a smartphone app made participants feel guilty, especially in a situation when exercise was not possible [55]. 

The long duration of battery time is another important facilitator; along with being comfortable, light, and user-friendly, wearable technology needs to be power efficient. We observed that majority of the accelerometers studies monitored physical activity for seven days which has been shown to enhance the robustness of the physical activity measurements and is a manageable length of time for most patients to wear the sensor, resulting in sufficient compliance [105]. Although sensor miniaturization has made it possible to measure for days or weeks without the need for large and bulky batteries or base stations, identified research trends reveal that research on battery technology lags compared with research on other wearable system components, implying that energy efficacy and efficiency remain an important design concern [106]. 

### 4.3. Recommendations for Future Research

The translation to a long-term commitment to wearable technology for health monitoring and management in a real-world setting requires clear use scenarios, valuable feedback, and constructive recommendations. The findings of this review provide guidance for planning and designing research that aims to utilize wearable technology in people with knee OA for these purposes. 

First, we identified gaps in the literature in terms of reporting. We recommend that authors collect and report data on multiple aspects of user experience, such as adherence to using wearable technology, adverse events, and acceptability. In addition, reporting of technical specifications of wearable technology such as weight, size, price, sampling frequency, and battery time was missing in most studies but is critical for further development of such technologies. Moreover, due to inconsistencies in how physical activity data is processed, publishing raw data sets as Appendix A should be standard practice. In accelerometers, for instance, an important feature might be the possibility to extract raw data that can be analyzed independent of the manufacturers’ algorithms or re-analyzed retrospectively if new algorithms are developed. Likewise, we found that no studies provided any open software to accompany their articles. Open-source software repositories (i.e., GitHub) can support cost-effective replication, advancement, and growth of software for clinical research [107]. Although limited, such examples are already present in the field of wearable technology [108,109]. Finally, a set of standard outcome measures and a testing methodology should be established in wearable technology for health outcome assessment in people with knee OA. We extracted close to 90 different data metrics, and most studies did not report the methodology of using wearable technology in participants’ real-world settings. Within optical gait analysis, the standard testing methodology is well established in knee arthroplasty research [110]. Similar progress is required in research on the utilization of wearable technology in a real-world setting in knee OA. 

Second, we observe significant gaps in the capability of wearable technology to collect other health outcomes than physical activity-based metrics. Although knee OA is considered to be partially driven by abnormal biomechanics [111], there is limited research in terms of using wearable technology for gait analysis in a real-world setting. A combination of accelerometers, gyroscopes, and magnetometers (IMU devices) may provide added benefits to measuring more complex metrics related to the biomechanics of gait, such as kinematics or kinetics, and thus provide insights into the quality of gait in various situations that would not be detected within constrains of in-lab setting [112]. IMUs have however significant power and data storage requirements [113], which requires a level of active interaction by the user for battery charging and initiation of data collection and transfer. In addition, measuring more complex metrics would require the integration of valid activity classification and event detection algorithms, which are usually studied in isolation but rarely incorporated together in a single system [108]. Hence, due to these technical and logistical difficulties, longitudinal assessment via IMUs is currently lacking. Nevertheless, such examples are already present in other fields [114,115] and can serve as a basis for further development of such devices for the assessment of biomechanics in people with knee OA in a real-world setting.

Finally, studies using machine learning and AI for delivering personalized support based on the data from wearable technology were not present among included studies, despite progress in AI-enable healthcare delivery [116]. The large volume of heterogeneous data types collected using wearable technology is beyond the abilities of commonly used data processing techniques. High-performance computing such as AI, machine learning, or deep neural networks permits the efficient processing of large volumes of data, but future studies need to consider how such an approach would transform collected data into manageable and useful information for people with knee OA. Current commercial wearable technologies lack relevance for many potential users, presenting an additional burden [117]. To gain wider consumer preference, the data collected with wearable technology has to be fitted into specific contexts, offering the needed insights and advice. 

### 4.4. Limitations

First, we did not conduct the assessment of the risk of bias and evidence strength. This is, however, not mandatory in scoping reviews [17] and, given the variety of designs of the included studies, would be out of the scope of this review. Second, it is questionable whether the results of this study can be extrapolated to people with a severe stage of knee OA as the included studies recruited relatively young people with knee OA (62–66 years of age). Finally, we did not include studies that investigated wearable technology in people with knee OA scheduled for or after knee joint replacement surgery. 

## 5. Conclusions

The research utilization of wearable technology in a real-world setting in people with knee OA is increasing, with the use of accelerometers for physical activity assessment being the most prevalent mode of use. There is a gap in research regarding the use of wearable technology for patient self-treatment and for information-sharing. To fully realize the clinical potential of wearable technology for people with knee OA, this review highlights the need for more research employing wearable technology for information sharing and treatment, increased inter-study consistency through standardization and improved reporting, and increased representation of vulnerable populations.

## Figures and Tables

**Figure 1 jcm-10-05645-f001:**
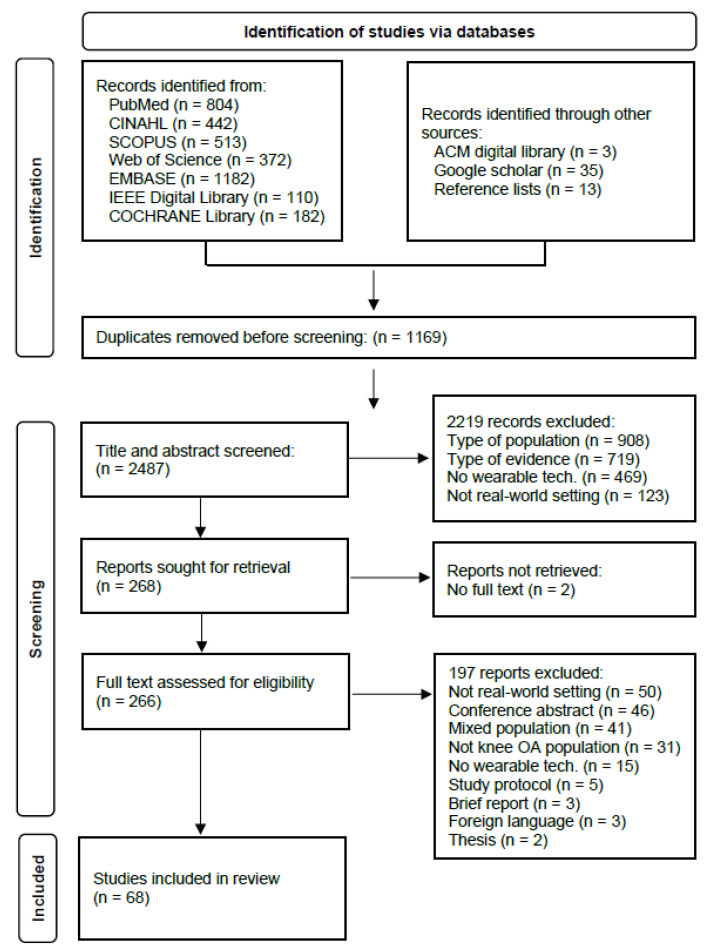
Flow diagram of the study selection process.

**Figure 2 jcm-10-05645-f002:**
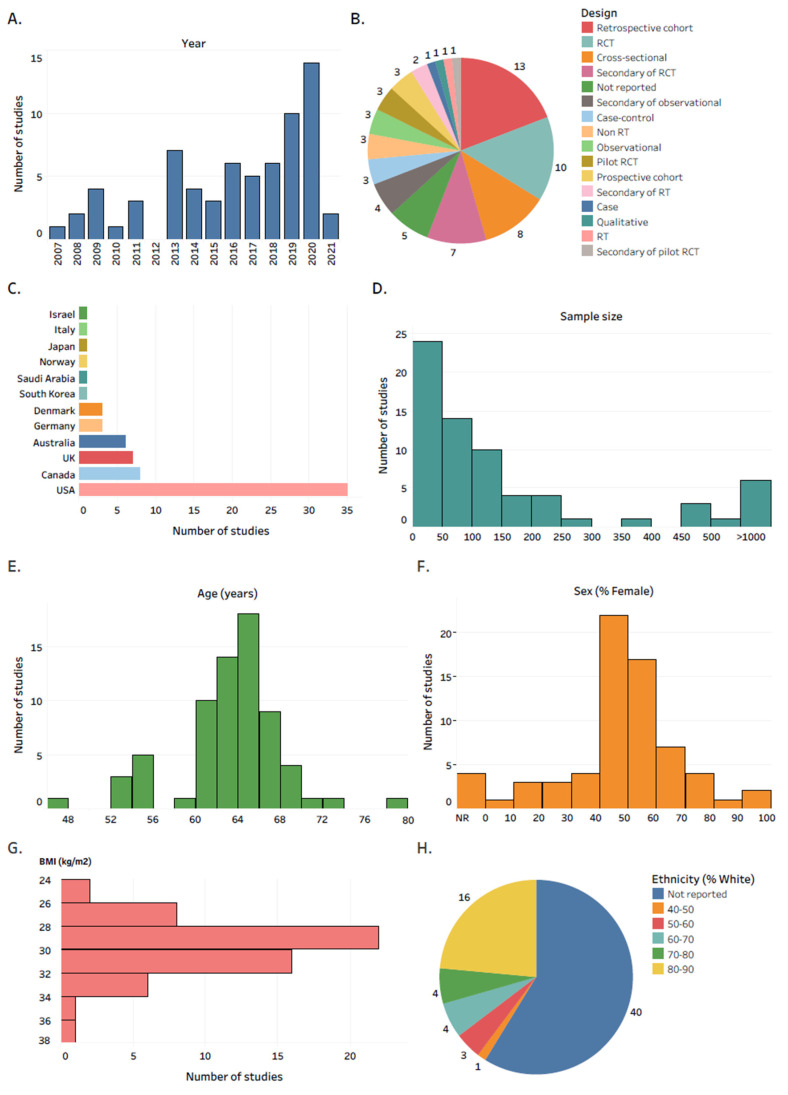
General study and population details. The number of studies by: (**A**) year of publication; (**B**) study design; (**C**) country; (**D**) sample size; (**E**) mean age; (**F**) percentage of female participants; (**G**) mean BMI; (**H**) percentage of participants of white ethnic background. Abbreviations: RCT, randomized controlled trial; RT, randomized trial; BMI, body mass index; NR, not reported. Interactive version of the figure with all the study references is located here: https://public.tableau.com/app/profile/tomasz.cudejko/viz/Figure2_16288559308710/Figure2 (accessed on 19 October 2021).

**Figure 3 jcm-10-05645-f003:**
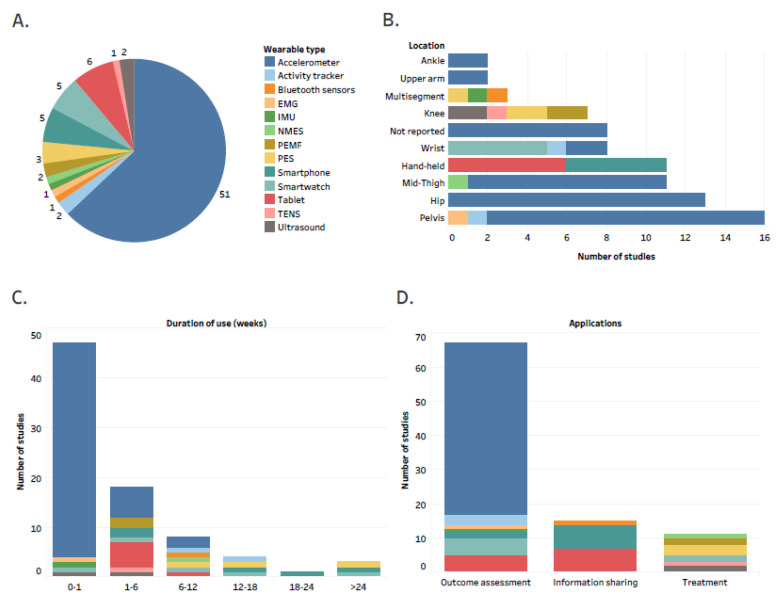
Details regarding type and applications of wearable technology; the number of studies by: (**A**) type of wearable technology; (**B**) location of body attachment stratified by type of wearable device; (**C**) duration of wearable use stratified by type of wearable device; (**D**) applications of wearable technology stratified by type of wearable device; Abbreviations: PES, pulsed electrical stimulation device; PEMF, pulsed electromagnetic field device; EMG, electromyography; IMU, inertial measurement unit; NMES, neuromuscular electric stimulation device; TENS, transcutaneous electrical nerve stimulation device; Interactive version of the figure with all the study references is located here: https://public.tableau.com/app/profile/tomasz.cudejko/viz/Figure3_16324888236280/Figure3 (accessed on 19 October 2021).

**Figure 4 jcm-10-05645-f004:**
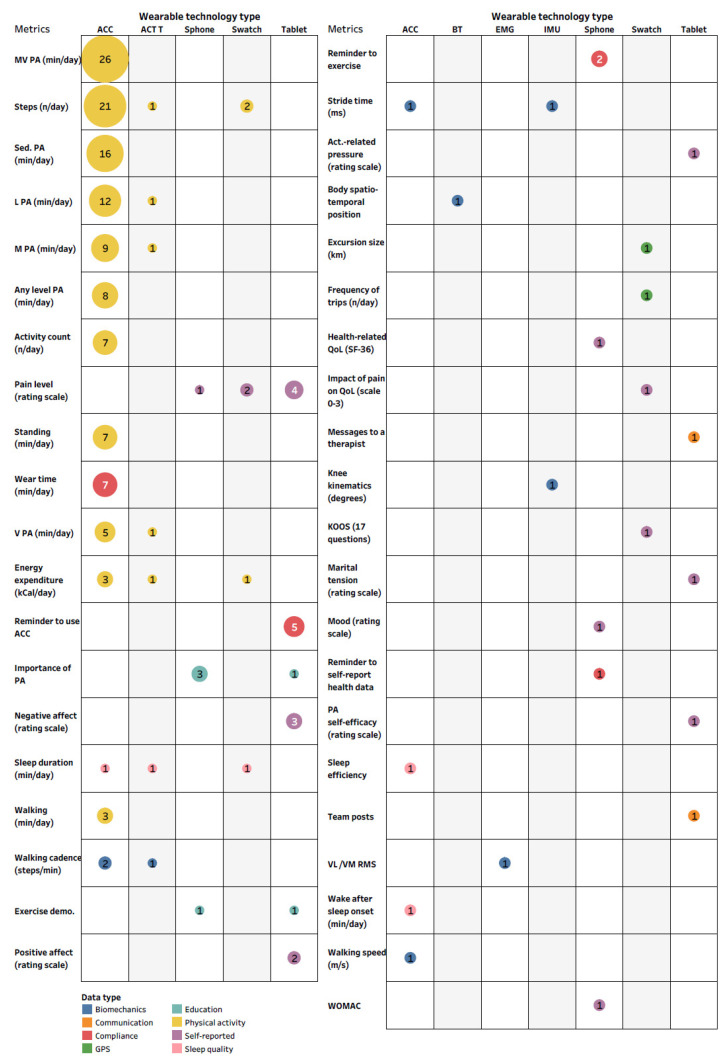
The number of studies by the specific data metrics recorded/transferred from/to the wearer stratified by data type and organized according to the type of wearable technology. Abbreviations: ACC, accelerometer; ACT T, activity tracker; Sphone, smartphone; Swatch, smartwatch; BT, Bluetooth sensors; EMG, electromyography; IMU, inertial measurement unit; PA, physical activity; MV, moderate to vigorous; L, light; Sed., sedentary; demo., demonstration; QoL, quality of life; KOOS, knee osteoarthritis outcome score; VL, vastus lateralis; VM, vastus medialis; RMS, root mean square; WOMAC - The Western Ontario and McMaster Universities Arthritis Index; Interactive version of the figure with all reported metrics and with all the study references is located here: https://public.tableau.com/app/profile/tomasz.cudejko/viz/Figure4_Full/Figure4_Full (accessed on 19 October 2021).

**Figure 5 jcm-10-05645-f005:**
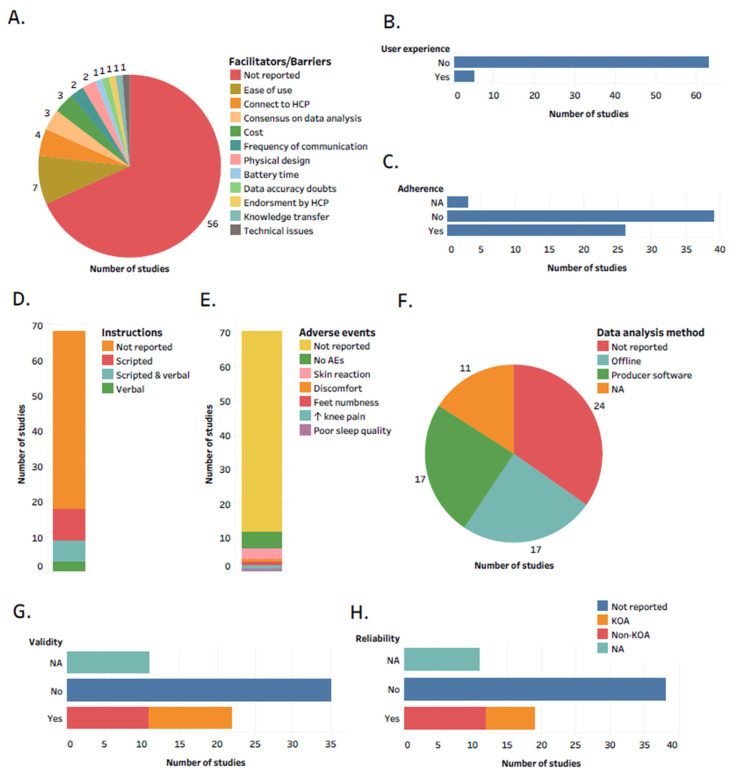
Details regarding facilitators and barriers to the feasibility of using wearable technology in a real-world setting in people with knee OA. The number of studies reported: (**A**) facilitators and barriers; (**B**) user experience; (**C**) adherence; (**D**) instructions provided to participants; (**E**) adverse events; (**F**) how data generated from wearable technology was analyzed; (**G**) validity; (**H**) reliability. Abbreviations: HCP, health care professional; NA, not applicable. KOA, knee osteoarthritis; AEs, adverse events; No, not reported; Yes, reported. Interactive version of the figure with all the study references is located here: https://public.tableau.com/app/profile/tomasz.cudejko/viz/Figure5_16328264416900/Figure5 (accessed on 19 October 2021).

## Data Availability

Full dataset is available in the Appendix A.

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
