# Peer review of "Applications of Wearable Technology in a Real-Life Setting in People with Knee Osteoarthritis: A Systematic Scoping Review"

_jcm, 2021, doi:10.3390/jcm10235645_

Round 1
Reviewer 1 Report
Dear Authors,
First, I would like to congratulate you for your efforts in such an important topic. As we are facing new challenges in clinical setting due to the epidemic, it has been crucial to implement technological improvements in our practices and to accelerate this process. With the utilization of wearable technologies it is undoubtedly possible to promote a continuous and a more effective treatment and rehabilitation program. From this point of view, i find your scoping review an enlightening work which gives an overview of the current literature on utilization of wearable technology in a real-world setting in people with knee osteoarthritis.
I have carefully go throughout the review. It has been well-organised and well written. However, I think at least two points needed further clarification. Here you can find them;
1-) In the pre-registered protocol of the scoping review, it has been mentioned that “methodology of the review has been done according to the scoping review methodology developed by Arksey and O’Malley who describe six stages of scoping review conduct: (1) identifying the research question (RQ), (2) identifying relevant studies, (3) selecting studies, (4) charting the data, (5) collating, summarizing and reporting results and (6) consulting with relevant stakeholders.” However, in the manuscript same methodology was described and applied with 5 stages in which excludes the last stage: consulting with the stake holders. (Please see line 79) What is the reason for this contradiction between the protocol and manuscript?
2-) Reviewing process ended up with inclusion of 68 articles. However, none of those articles were cited in the results and relevant parts throughout the paper. Is there any specific reason to not cite the included articles? If not, please cite the articles. Here are some sentences can be cited with relevant articles. Please, give citation to similar statements throughout paper. Line 161, 168, 170, 174, 183, 184, 185, 214, 219, 297, 313, 314
To exemplify (Line 313, 314): “We found 11 studies that used wearable technologies, such us ultrasound (REFERENCES), PES (REFERENCES), PEMF (REFERENCES), and or TENS (REFERENCES), for treatment of people with knee OA in a real-world setting.”
Author Response
Dear Reviewer,
Thank you for your review. Please see the attachment for our response to the comments.
Kind regards
Tomasz Cudejko
Reviewer 2 Report
This paper is well written and organized logically, which make it quite readable. I recommend that the paper may be accepted for publication in present form.
Author Response
We thank the Reviewer for reviewing our manuscript and the positive feedback.
Reviewer 3 Report
Title: please consider “real-life” instead of “real-world”
Abstract should not start with a statement on COVID-19 which is only presented as secondary reason to use wearable technology.
Keywords - osteoarthritis and arthritis are two very different diseases.
Row 24 - osteoarthritis is not a rheumatic disease. Please provide accurate medical information when refer to a certain pathology. Any kind of rheumatism affects one way or another the osteoarticular system, therefore the information is redundant. To sustain the statement that the knee is the most affected joint your reference from Lancet is not appropriate. Please provide another or multiple references for this statement
Row 26 - the anatomopathological description of cartilage lesions together with signs and symptoms described are not specific for osteoarthritis, even if they are present in OA
Please provide reference for your statement in row 30, since obesity is not a determinant factor for OA.
Row 44 - please rephrase or explain the meaning of “precision medicine”
Row 49 - maybe more appropriate is “elderly people”, not “older people”, please consider rephrasing
Row 58 - “analysis of walking gait”, the medical terminology is “gait analysis”
Row 60 - “following knee joint replacement surgery specifically”, please rephrase or explain the meaning of your statement. A total/partial knee arthroplasty can be only specific surgery.
Row 73-77 - this phrase should be either in materials and methods or these questions should be answered/discussed in introduction as presentation of what wearable technology is used for.
Throughout introduction there is no presentation about how wearable devices are used in patients with osteoarthritis. In row 38 there is a nice definition of wearable devices followed by a presentation of AI, but in patients with OA for sure not the applications in fitness industry are used.
Inclusion criteria are well presented.
Row 106 - patients “scheduled for/or after any form of joint surgical procedure” were excluded. This is an important statement, showing that this review addresses studies including patients with early osteoarthritis. Among the important benefits of wearable technologies is included the functional (subjective) postoperative evaluation of patients for assessment of the quality of life, therefore I consider that this aspect should be mentioned.
Row 106 - “inflammatory type of OA” - any rheumatic disease includes an inflammatory process, even in degenerative OA there is a reduced inflammatory component, please check your statements throughout the entire document for these false statements.
Appendix B is very well used. However, please explain why you have used different keywords for the search in different databases, instead of using same words. Different searches may provide different results.
Row 121 - from your statement I understand that the inclusion and exclusion criteria are decided after you have started this study. This means that you could subjectively change those criteria to fulfill some of your expectations. Please explain or rephrase.
Row 143 - using only excel tools for statistical analysis may not be the best way to provide statistical relevant information. Please explain this option
Very well used figure 1 for providing information
Figure 2 - I found almost no discussion about the data presented in this figure. What is the relevance of number of studies involving females? Are there different results considering ethnicity? And I refer here to analyzing the results of this study compared with literature.
Figure 3 - Accelerometers and gyroscopes are also used by smartwatches and smartphones. If we consider smartphone and tablets as tools for uploading collected data, I consider that this category should be well differentiated from the devices used for acquiring information. There is a mix of detection devices, transmission data devices and treatment/stimulation devices. I consider these categories should be better defined in figure 3A, similar to what you did in figure 3D.
Row 189 please specify if possible if you refer to natural outcome or treatment outcome. Monitoring the “natural” evolution of a disease it different from evaluation of the treatment clinical and functional outcome.
Figure 4 - the scores evaluated with wearable devices (like KOOS, WOMAC, SF-36, etc.) includes pain, walking, and other similar information.
Figure 5 provide important information, very well structured
Row 242 - The discussion section should start with the most important finding of this study. I recommend adding more relevant information in the first statement of this paragraph or rephrase it.
Row 250 - “very recent growth in the use of wearable technology in a real-world setting in people with knee OA “- please provide reference for this statement
Row 254 - inclusion criteria were OA without surgical indication for treatment. This excludes per se patients with advanced OA for this study. Of course these results questions the extrapolation of this analysis in patients with advanced OA. Therefore, I consider that your statement should be in section: limitation of the study/wearable technology. Same comment for row 307.
Row 325 - for this paragraph you should provide reference since your statements are not resulting from this study data.
Row 341 - I suggest to emphasize the difference of costs related to evaluation vs treatment devices.
Row 354 - “We additionally show that 50% of studies determined validity and reliability on non-knee OA population.”- why are these studies included in this analyze? Please explain.
I find very appropriate the statements provided in paragraph started in row 389
Row 438 - the author should refer to the limitations of this analyze, not to data that were not included in this study design, as advanced OA or radiology in OA. This is the first mention of imagistic investigations for assessment of OA in this paper. There was no discussion about criteria for the degree of OA. Therefore, these statements should be better placed in future research (for an analyze, not for limitations of wearable devices).
Row 444 - this phrase should be in introduction. This is not a limitation of your study.
Row 447 - “The utilization of wearable technology in a real-world setting in people with knee OA is increasing,”- this is not a conclusion of your study, this statement does not result from data collected in your study
Reference 5 is a webpage with reports, please refer to a specific document
Author Response
Dear Reviewer,
Thank you for your review. Please see the attachment for our response to the comments.
Kind regards,
Tomasz